# Long COVID Classification: Findings from a Clustering Analysis in the Predi-COVID Cohort Study

**DOI:** 10.3390/ijerph192316018

**Published:** 2022-11-30

**Authors:** Aurélie Fischer, Nolwenn Badier, Lu Zhang, Abir Elbéji, Paul Wilmes, Pauline Oustric, Charles Benoy, Markus Ollert, Guy Fagherazzi

**Affiliations:** 1Deep Digital Phenotyping Research Unit, Department of Population Health, Luxembourg Institute of Health, L-1445 Strassen, Luxembourg; 2Bioinformatics Platform, Luxembourg Institute of Health, L-1445 Strassen, Luxembourg; 3Luxembourg Centre for Systems Biomedicine (LCSB), University of Luxembourg, Campus Belval, L-4362 Esch-sur-Alzette, Luxembourg; 4Department of Life Sciences and Medicine, Faculty of Science, Technology and Medicine, University of Luxembourg, L-4367 Belvaux, Luxembourg; 5Association Après J20 COVID Long France, F-28110 Lucé, France; 6Centre Hospitalier Neuro-Psychiatrique, L-9002 Ettelbruck, Luxembourg; 7Psychiatric Hospital, University of Basel, 4002 Basel, Switzerland; 8Department of Infection and Immunity, Luxembourg Institute of Health, L-4354 Esch-sur-Alzette, Luxembourg; 9Department of Dermatology and Allergy Center, Odense Research Center for Anaphylaxis (ORCA), University of Southern Denmark, 5000 C Odense, Denmark

**Keywords:** clustering, COVID-19, Long COVID, disease severity

## Abstract

The increasing number of people living with Long COVID requires the development of more personalized care; currently, limited treatment options and rehabilitation programs adapted to the variety of Long COVID presentations are available. Our objective was to design an easy-to-use Long COVID classification to help stratify people with Long COVID. Individual characteristics and a detailed set of 62 self-reported persisting symptoms together with quality of life indexes 12 months after initial COVID-19 infection were collected in a cohort of SARS-CoV-2 infected people in Luxembourg. A hierarchical ascendant classification (HAC) was used to identify clusters of people. We identified three patterns of Long COVID symptoms with a gradient in disease severity. Cluster-Mild encompassed almost 50% of the study population and was composed of participants with less severe initial infection, fewer comorbidities, and fewer persisting symptoms (mean = 2.9). Cluster-Moderate was characterized by a mean of 11 persisting symptoms and poor sleep and respiratory quality of life. Compared to the other clusters, Cluster-Severe was characterized by a higher proportion of women and smokers with a higher number of Long COVID symptoms, in particular vascular, urinary, and skin symptoms. Our study evidenced that Long COVID can be stratified into three subcategories in terms of severity. If replicated in other populations, this simple classification will help clinicians improve the care of people with Long COVID.

## 1. Introduction

It is now estimated that a mean of 10 to 20% of the people infected by the SARS-CoV-2 experience persisting and fluctuating symptoms more than 12 weeks after the acute infection [1,2]. This syndrome has been called “Long COVID” by patients themselves and has a high impact on the quality of life of the affected people and, as a consequence, on the whole healthcare system. 

Long COVID has been defined by WHO as a condition that occurs 3 months after infection with SARS-CoV-2, with symptoms that last at least 2 months and cannot be explained by any other diagnosis [3], but this definition does not account for the substantial intragroup variability in the different presentations of Long COVID.

Many studies described Long COVID in post-hospitalization cohorts [4,5,6] and, with similar results, in population-based studies of less severe forms of COVID-19 [7,8]. The most commonly reported symptoms are fatigue, shortness of breath, and cognitive dysfunction, usually having a major impact on daily life [3,7,8]. Long COVID affects many organs with pulmonary, cardiac, thromboembolic, neurologic, and renal sequelae. However, their distribution and intensity in the general population are largely heterogeneous [9]. 

A one-size-fits-all care strategy for people with Long COVID is therefore not possible and a better understanding of the subforms of Long COVID would allow for developing personalized care for people with Long COVID or could be integrated as a screening tool for future clinical trials [10]. To date, few studies used clustering analysis to identify and characterize different Long COVID phenotypes [8,11,12].

In this study, we hypothesized that Long COVID can be stratified into different clinically relevant subgroups. We applied hierarchical clustering to study participants with Long COVID from the Predi-COVID cohort study to test this hypothesis.

## 2. Materials and Methods

### 2.1. Study Population

We used data from the Predi-COVID study, a prospective cohort study of persons in Luxembourg with a PCR-confirmed diagnosis of COVID-19. The study design and objectives have been published previously [13]. Participants were followed-up at 12 months with a self-reported questionnaire to update their general health status, persisting symptoms, and quality of life. The Predi-COVID study was approved in April 2020 by the National Research Ethics Committee of Luxembourg (study number 202003/07) and by the Luxembourg Ministry of Health as the authorizing body.

Individual characteristics, comorbidities, and initial symptoms were collected at inclusion during the Predi-COVID study. Initial COVID-19 disease severity (“Asymptomatic”, “Mild illness”, and “Moderate/severe illness”) has been previously assessed, as described elsewhere [14,15].

Persisting symptoms were collected using a list of 62 symptoms [10], further divided into 8 categories: ear/nose/throat symptoms, neurological and ocular symptoms, general symptoms, cardiorespiratory symptoms or diseases, gastrointestinal symptoms, vascular and ganglionic symptoms or diseases, urinary symptoms, and skin symptoms (see online Appendix A for the full list).

Sleep quality was assessed using the Pittsburgh Sleep Quality Index [16]. The respiratory quality of life was assessed with the VQ11 questionnaire (global score and 3 subscores) [17]. Finally, participants were asked whether they could envisage coping with their current health status in the long term (yes/no).

Inclusion criteria for our analysis were: adult participants with a complete 12-month questionnaire and baseline data available and who declared at least one persisting symptom.

### 2.2. Clustering and Statistical Analysis

The clustering was based on the following features: sociodemographic characteristics, initial classification of COVID-19 disease severity, comorbidities, symptoms at inclusion, and quality of life (see online Appendix A for the full list). 

A hierarchical ascendant classification (HAC) was used to construct clusters [8]. The optimal number of clusters was determined using the “elbow” method, which calculates the distortion depending on the number of clusters with the objective to maintain clinical interpretability and sufficient cluster size. The cluster stability was assessed with the Jaccard similarity index. A simple imputation was done for variables if they had less than 5% of missing data (using median for quantitative variables and main modality for categorical variables) and multiple imputations using the mice package from R otherwise. Data were described with numbers and percentages for categorical variables and with mean and standard deviation for numerical variables. We performed the analysis by using R software v4.1.2 [18] and generated the figures by using the ggplot2 R package [19]. 

## 3. Results

### 3.1. Population Study Characteristics

We initially included 545 participants between May 2020 and May 2021 with an available follow-up questionnaire 12 months after their primary infection. Participants with incomplete questionnaires were excluded (N = 54) as were participants aged less than 18 years (N = 1), and participants without any information about their study inclusion (N = 19) or about their initial COVID-19 severity classification (N = 3). Participants who did not experience any symptoms at 12 months were removed (N = 180). Finally, 288 participants were considered in the analysis (see online Appendix A).

Most of the overall study participants were female (59%) and not hospitalized at the time of COVID-19 (97%). The average age was 43 years (sd = 12) and 16% of the participants were smokers. One-third (33 %) of the participants had a moderate/severe form of the initial COVID-19. Sixty percent of the participants experienced poor sleep quality (PSQI total score > 5) and 28% had a poor respiratory quality of life (VQ11 global score > 22). Few participants had comorbidities prior to COVID-19 diagnosis (14%), and they had an average of 2.38 (sd = 0.33) comorbidities. Hypertension was the most frequent one (13%). At the time of inclusion, the most frequent symptoms were fatigue/malaise (47%), fever (34%), cough (33%), cephalea (27%), and rhinorrhea (26%). 

On average, participants declared eight symptoms (sd = 8) after 12 months. Most participants had general symptoms (80%), neurological and ocular symptoms (65%), and cardiorespiratory symptoms (55%). 

### 3.2. Clusters

Based on the elbow curve (see Figure 1), we determined the optimal cluster number to be three, which simultaneously allows good cluster stability (Cluster-Mild, Jaccard = 0.5707; Cluster-Moderate, Jaccard = 0.7556; and Cluster-Severe, Jaccard = 0.8297), clinical interpretability, and sufficient cluster size for each cluster.

We labeled them according to their distinguishing characteristics. The characteristics of the overall study population and of the three clusters are shown in Table 1.

Cluster-Mild contains 139 participants (48.26%). Compared with the overall study population, the initial disease severity was classified as moderate/severe for only 24% of the members of Cluster-Mild. Individuals in this cluster had a less impacted quality of life than the overall study population: only 7.9% declared that they could not envisage coping with their symptoms in the long term, 40% of them had poor sleep quality, and 5.8% had poor respiratory quality of life. Overall, participants in Cluster-Mild had fewer comorbidities (8.6%). At 12 months, participants declared fewer symptoms overall (mean number = 2.89, sd = 2.15). The symptoms were mostly grouped in the following categories: general symptoms (58%), neurological and ocular symptoms (37%), and cardiorespiratory symptoms or diseases (24%).

Cluster-Moderate contains 106 participants (36.81%). Compared with the overall study population, members were slightly more frequently female (62%) and presented more frequently a moderate/severe form of the initial illness (39%). Quality of life was more impacted with 23% of Cluster-Moderate declaring that they could not envisage coping with their symptoms in the long term, 78% of them having a poor sleep quality, and 48% having a poor respiratory quality of life. Comorbidities were similar in Cluster-Moderate and in the overall study population but participants declared a higher number of symptoms at 12 months (mean = 11.5, sd = 5.7). All participants had general symptoms (100%), and a large majority also had neurological and ocular symptoms (95%) and cardiorespiratory symptoms or diseases (82%). Most participants also had ENT symptoms (61%).

Cluster-Severe contains 43 participants (14.93%). Compared with the overall study population, members were mostly females (72%). Participants were more frequently smokers (33%) and 47% had an initial moderate/severe acute illness. Similar to Cluster-Moderate, the quality of life in Cluster-Severe was highly impacted with 84% of them having poor sleep quality and 51% having a poor respiratory quality of life. Overall, participants in Cluster-Severe presented more comorbidities at inclusion (28%), hypertension being the most frequent one (28%). At 12 months, participants had a high number of symptoms (mean = 18, sd = 9). The presentation of symptoms was similar to Cluster-Moderate for general, neurological, and cardiorespiratory symptoms: all participants had general symptoms (100%), 84% had neurological and ocular symptoms or diseases, and 91% had cardiorespiratory symptoms or diseases. High frequencies of vascular, skin, and urinary symptoms (86%, 86%, and 33%, respectively) characterize Cluster-Severe.

The symptom distribution by symptom categories in the three clusters is represented in Figure 2, which shows the differences among the clusters.

## 4. Discussion

In this study, we identified three clusters of Long COVID in people with persisting symptoms 12 months after acute infection with a clear gradient in Long COVID severity. Cluster-Mild represented almost half of the study population and was composed of participants with less severe initial infection, fewer comorbidities, and with few persisting symptoms (mean = 2.9), mainly in the general, neurological, or cardiorespiratory categories. Individuals in Cluster-Moderate declared a mean of 11.5 persisting symptoms and had poor quality of sleep and of respiratory quality of life. Cluster-Severe was characterized by a higher proportion of women, smokers, and a higher number of pre-existing comorbidities than in Clusters-Mild and Clusters-Moderate. Strikingly, participants from Cluster-Severe declared more persisting symptoms in total than those from Cluster-Moderate (mean = 18), with a similar pattern of general, neurological, and cardiorespiratory symptoms, but is distinct by higher occurrences of vascular, urinary, and skin symptoms. 

General symptoms were predominant in all three clusters. This is in line with previous findings showing that general symptoms were the most frequently reported symptoms in people with persisting symptoms at 12 months, with a predominance of fatigue (34.3%), irritability (18%), anxiety (15.9%), muscle or joint pain in the lower limbs (15.6%), and back pain (14.9%) [10].

Few studies investigated clustering analysis of Long COVID patients. Kenny et al. applied similar clustering methods to a prospective cohort of 233 COVID-19-infected patients with ongoing symptoms at least 4 weeks after acute infection and also described three clusters: the largest constituted by participants with a lower number of persisting symptoms (mean = 2) and two characterized by a higher number of persisting symptoms (mean = 4 and 6) and more functional impairments. As in our study, the distribution of persisting symptoms was different between the two most severe clusters, with one cluster grouping cardiorespiratory and general symptoms, and the other one with a predominance of pain-related symptoms. The time and method of symptom evaluation were different as it was done in person during a visit to a clinic and the median time of symptom duration was 18 weeks [12]. Another study identified three different clusters among a cohort of 1969 post-hospitalized COVID-19 patients in Spain [11]: one cluster grouped patients with fewer comorbidities and symptoms at the hospital inclusion, less persisting symptoms, and had a preserved quality of life, and the other two clusters were constituted of patients with more pre-existing comorbidities, a higher number of symptoms during the acute phase, a higher number of persisting symptoms, and greater impact on quality of life (higher level of anxiety and altered sleep quality). One cluster was also characterized by respiratory symptoms (dyspnea at rest, 73.4%) and particularly high limitations in daily activities (92.1% for social activities and 93.3% for instrumental daily activities). The overall number of symptoms in each cluster was lower than in our clusters because their clustering also included participants without persisting symptoms. 

Another study conducted in the United Kingdom in 2022 also described groups of people with Long COVID. More participants (N = 2550) were recruited, via an online survey, with a mean duration of illness of 7.2 months (sd = 1.8). The mean age was similar to our participants, as was the greater presence of women and comorbidities. The most common first symptoms (fatigue, headache, chest pain, shortness of breath, and cough), persistent symptoms (fatigue, cognitive dysfunction, chest pain, shortness of breath, headache, and muscle pain), number of symptoms experienced, and organ systems affected, were also similar. Participants were asked to report the presence or absence of 35 symptoms, and two groups were identified. The first group (88.8%) had mainly cardiopulmonary, cognitive, and fatigue symptoms and the second group had more multisystem symptoms [8], which aligns relatively well with our findings.

Reese et al. applied an adapted Phenomizer algorithm to classify patients with Long COVID, based on the ICD-10 diagnosis code U09.9 for post-COVID-19 condition, and identified six clusters [20]. Although the clustering method was different and based on medical records data, this study also identified two “severe” clusters with more pre-existing comorbidities, an increased initial illness, and a wide range of Long COVID symptoms. 

The larger representation of women in the most severe cluster is consistent with findings from other studies [8,12]. 

Finally, despite different analysis time points, similar results were found in these different studies, which confirm that our findings are relevant despite the fluctuating character of Long COVID.

### Strengths and Limitations

This study has several strengths. First, a large list of 62 symptoms was considered, distributed in eight categories that cover the complex symptomatology of Long COVID. Participants with different forms of initial illness severity were represented. All participants had a documented initial COVID-19 infection, confirmed by a PCR test and their symptoms were assessed 12 months after acute infection. 

This study also has some limitations. The analyses were done on a moderate sample size and, as in any selected study population, results may not be directly extrapolated to all people with Long COVID. External validation in a larger population would be of the highest interest to confirm these results. Information on pre-existing symptoms before COVID-19 infection was missing and symptoms were self-reported, which could lead to bias in estimating the number of persisting symptoms attributable to COVID-19. However, this may not affect the main message of our findings. The participants in the present study were included before the Omicron wave; thus, we cannot ensure that our results can be extended to Long COVID following infection by the Omicron variant. Recent studies demonstrated that infection by Omicron variants leads to a 24 to 50% risk reduction of developing Long COVID; however, there were no differences in the distribution of Long COVID symptoms and the risk of neurological and psychiatric sequelae remains the same after infection by Omicron [21,22,23].

## 5. Conclusions

Our study highlighted three clinically relevant subgroups of people with Long COVID of increasing severity, but also with different patterns of symptoms. Such stratification of Long COVID will help healthcare professionals improve the triage and care of people with Long COVID.

## Figures and Tables

**Figure 1 ijerph-19-16018-f001:**
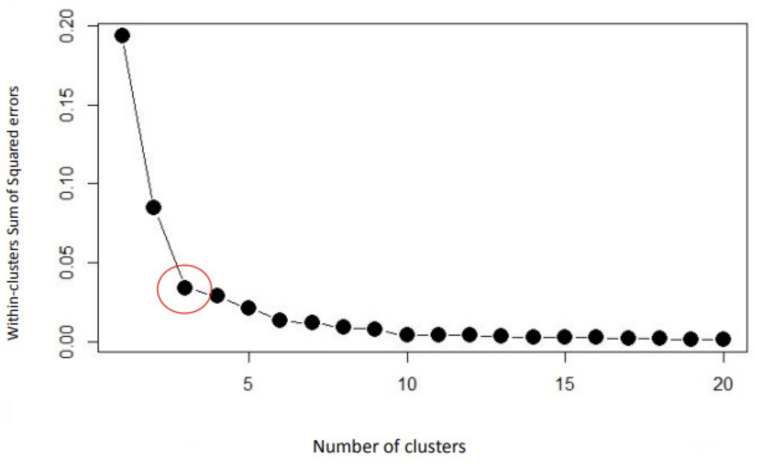
Determination of optimal cluster number. The optimal number is visualized by the inflection point that corresponds to three clusters, as shown by the red circle.

**Figure 2 ijerph-19-16018-f002:**
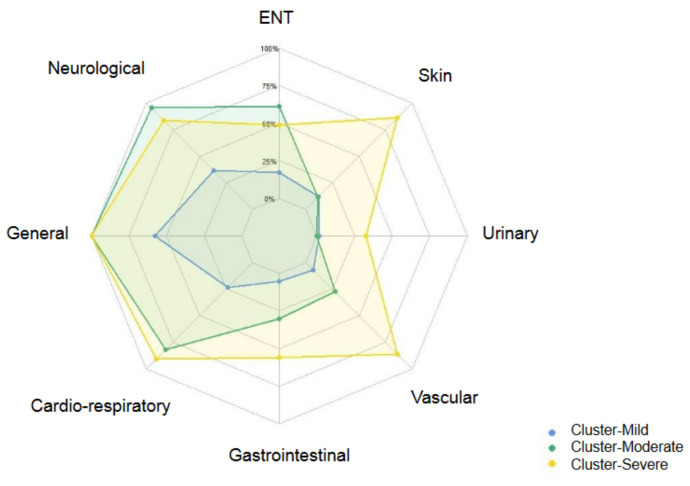
Distribution of Long COVID symptoms (in %) by symptom categories in the three clusters.

**Table 1 ijerph-19-16018-t001:** Participants’ characteristics in the overall study population and by cluster.

		Overall Population N = 288	Cluster-Mild N = 139 (48.26%)	Cluster-Moderate N = 106 (36.81%)	Cluster-Severe N = 43 (14.93%)	*p*-Value *
Sociodemographic Characteristics and Initial Severity Classification	Female N (%)	170 (59%)	73 (53%)	66 (62%)	31 (72%)	0.053
	Age (Years)	43 ± 12	42 ± 12	43 ± 12	45 ± 14	0.360
	Body Mass Index (kg/m2)	26.4 ± 5.5	25.8 ± 5.1	27.0 ± 5.8	26.7 ± 5.7	0.224
	Smoker N (%)	45 (16%)	16 (12%)	15 (14%)	14 (33%)	0.027
	Moderate/severe illness N (%)	95 (33%)	34 (24%)	41 (39%)	20 (47%)	0.015
Comorbidities	At least one comorbidity N (%)	40 (14%)	12 (8.6%)	16 (15%)	12 (28%)	0.007
	Number of comorbidities Mean (SD)	2.38 ± 0.33	2.37 ± 0.25	2.34 ± 0.16	2.48 ± 0.68	0.001
	Hypertension N (%)	38 (13%)	14 (10%)	12 (11%)	12 (28%)	0.015
	Cardiac diseases N (%)	11 (3.8%)	3 (2.2%)	6 (5.7%)	2 (4.7%)	0.311
	Asthma N (%)	14 (4.9%)	4 (2.9%)	8 (7.5%)	2 (4.7%)	0.200
	Diabetes N (%)	13 (4.5%)	3 (2.2%)	4 (3.8%)	6 (14%)	0.009
Symptoms at inclusion N (%)	Fever	98 (34%)	45 (32%)	36 (34%)	17 (40%)	0.688
	Cough	96 (33%)	41 (29%)	38 (36%)	17 (40%)	0.362
	Cough sputum	27 (9.4%)	11 (7.9%)	9 (8.5%)	7 (16%)	0.279
	Sore throat	50 (17%)	17 (12%)	24 (23%)	9 (21%)	0.076
	Rhinorrhea	76 (26%)	35 (25%)	31 (29%)	10 (23%)	0.708
	Earache	22 (7.6%)	8 (5.8%)	10 (9.4%)	4 (9.3%)	0.490
	Chest pain	19 (6.6%)	4 (2.9%)	11 (10%)	4 (9.3%)	0.036
	Myalgia	51 (18%)	11 (7.9%)	28 (26%)	12 (28%)	<0.001
	Arthralgia	25 (8.7%)	4 (2.9%)	14 (13%)	7 (16%)	0.001
	Fatigue	136 (47%)	47 (34%)	60 (57%)	29 (67%)	<0.001
	Dyspnea	33 (11%)	10 (7.2%)	16 (15%)	7 (16%)	0.067
	Cephalea	77 (27%)	27 (19%)	36 (34%)	14 (33%)	0.022
	Abdominal pain	14 (4.9%)	4 (2.9%)	3 (2.8%)	7 (16%)	0.004
	Nausea	13 (4.5%)	5 (3.6%)	4 (3.8%)	4 (9.3%)	0.289
	Diarrhea	20 (6.9%)	5 (3.6%)	8 (7.5%)	7 (16%)	0.019
Persisting symptoms at 12 months N (%)	Ear Nose Throat (ENT) symptoms	110 (38%)	24 (17%)	65 (61%)	21 (49%)	<0.001
	Neurological symptoms	188 (65%)	51 (37%)	101 (95%)	36 (84%)	<0.001
	General symptoms	229 (80%)	80 (58%)	106 (100%)	43 (100%)	<0.001
	Cardiorespiratory symptoms	159 (55%)	33 (24%)	87 (82%)	39 (91%)	<0.001
	Gastrointestinal symptoms	63 (22%)	7 (5.0%)	32 (30%)	24 (56%)	<0.001
	Vascular symptoms	76 (26%)	10 (7.2%)	29 (27%)	37 (86%)	<0.001
	Urinary symptoms	16 (5.6%)	2 (1.4%)	0 (0%)	14 (33%)	<0.001
	Skin symptoms	66 (23%)	17 (12%)	12 (11%)	37 (86%)	<0.001
Number of persisting symptoms at 12 months Mean (SD)	Total number of symptoms	8 ± 8	2.89 ± 2.15	11.5 ± 5.7	18 ± 9	<0.001
	Number ENT symptoms	0.70 ± 1.11	0.25 ± 0.63	1.12 ± 1.24	1.09 ± 1.44	0.079
	Number neurological symptoms	2.12 ± 2.28	0.72 ± 1.27	3.27 ± 2.07	3.79 ± 2.63	<0.001
	Number general symptoms	3.02 ± 2.86	1.19 ± 1.48	4.04 ± 2.30	6.44 ± 3.13	<0.001
	Number cardiorespiratory symptoms	1.36 ± 1.72	0.42 ± 0.92	2.02 ± 1.65	2.81 ± 2.11	0.002
	Number gastrointestinal symptoms	0.39 ± 0.87	0.079 ± 0.382	0.48 ± 0.86	1.19 ± 1.35	0.010
	Number vascular symptoms	0.39 ± 0.75	0.09 ± 0.33	0.41 ± 0.73	1.35 ± 0.95	0.356
	Number urinary symptoms	0.07 ± 0.32	0.01 ± 0.11	0.00 ± 0.00	0.44 ± 0.70	0.610
	Number skin symptoms	0.27 ± 0.54	0.14 ± 0.38	0.13 ± 0.39	1.05 ± 0.62	0.570
Quality of life N (%)	Could not envisage coping with symptoms long term	45 (16%)	11 (7.9%)	24 (23%)	10 (23%)	0.002
	Poor sleep #	239 (83%)	102 (73%)	99 (93%)	38 (88%)	<0.001
	Altered respiratory quality of life at 1 year	81 (28%)	8 (5.8%)	51 (48%)	22 (51%)	<0.001

Sleep quality was assessed using the PSQI questionnaire. A categorical variable was generated using the PSQI score: ^#^ poor sleep was defined as PSQI total score > 5. The respiratory quality of life was assessed using the VQ11 questionnaire, initially developed for COPD patients. One global score and 3 subscores (functional, psychological, and relational) were calculated as described elsewhere and categorical variables were generated. Altered respiratory quality of life was defined as VQ11 global score > 22. * *p*-values are determined using the ANOVA significant difference test for continuous variables (age and BMI) and Fisher’s exact test for categorical variables.

## Data Availability

Data are available from the corresponding author upon reasonable request.

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
