# Peer review of "Long COVID Classification: Findings from a Clustering Analysis in the Predi-COVID Cohort Study"

_ijerph, 2022, doi:10.3390/ijerph192316018_

Round 1

Reviewer 1 Report

Overall I found this manuscript well written and easy to read. The argument was well researched and presented clearly.

The concept of clustering patients with ongoing symptoms following covid19 infection is not new but this manuscript adds to the growing body of literature that suggests patients can fall into one of three broad categories. This manuscript is particularly clear in the presentation of this approach. As such, this article is useful but also somewhat limited.

The claim that this mild/moderate/sever categorisation is related to "personalised medicine" overstates the data as presented. Whilst it appears to be true that there is a cohort of patients with severe ongoing symptoms that are fewer in number than those with milder long covid, the severe symptoms are still quite varied. Treatment options for these patients would therefore also differ quite widely. 

It is noticeable that 'general' symptoms is the largest grouping for all categories. Since mild/moderate/severe classification for long covid is now quite a common approach, there may be value in the authors exploring these 'general' symptoms in more detail and providing a more thorough analysis of this group in all categories.

As a minor point I question the classification of 'blue fingers' as a skin symptom. Without additional context this would seem to be an indication of a problem with blood circulation.

Author Response

Overall I found this manuscript well written and easy to read. The argument was well researched and presented clearly.

The concept of clustering patients with ongoing symptoms following covid19 infection is not new but this manuscript adds to the growing body of literature that suggests patients can fall into one of three broad categories. This manuscript is particularly clear in the presentation of this approach. As such, this article is useful but also somewhat limited.

Point 1: The claim that this mild/moderate/sever categorisation is related to "personalised medicine" overstates the data as presented. Whilst it appears to be true that there is a cohort of patients with severe ongoing symptoms that are fewer in number than those with milder long covid, the severe symptoms are still quite varied. Treatment options for these patients would therefore also differ quite widely. 

Our response to point 1: We thank reviewer 1 for this comment. We agree that the term ”Personalized medicine” might be too strong. However, we are convinced that this classification will participate in personalizing the treatments for Long COVID patients, in combination with other relevant clinical information. Following the Reviewer’s recommendation, we have decided to use the term “stratification” to better reflect the potential of our approach and we rephrased a bit our conclusion. 

Point 2: It is noticeable that 'general' symptoms is the largest grouping for all categories. Since mild/moderate/severe classification for long covid is now quite a common approach, there may be value in the authors exploring these 'general' symptoms in more detail and providing a more thorough analysis of this group in all categories.

Our response to point 2: We fully agree with Reviewer 1. We have previously observed (Fischer et al, 2022. https://doi.org/10.1093/ofid/ofac397) that “General symptoms” were the most frequently reported symptoms in people with persisting symptoms at 12 months and their distribution was: fatigue (34.3%), irritability (18%), anxiety (15.9%), muscle or joint pain in the lower limb (15.6%), back pain (14.9%). 

However, we did not observe any specific trend in this category compared to the other symptom categories when looking across the clusters, beyond what is already mentioned in the current manuscript. 

We have added this point in the discussion, lines 254-258.  

Point 3: As a minor point I question the classification of 'blue fingers' as a skin symptom. Without additional context this would seem to be an indication of a problem with blood circulation.

Our response to point 3: We thank Reviewer 1 for this remark. “Blue fingers” linked to COVID-19 have been from the start described as skin rashes, even though the physiological explanation is indeed probably linked to blood vessel disorders in the fingers. This symptom is generally considered as a dermatological symptom,  that’s why we categorized it in the “Skin symptoms” category. This will also further help the comparison between our results and other publications on Long COVID symptomatology.

Reviewer 2 Report

It is an interesting report highlighting "Long COVID" symptoms. However, as indicated by the authors that The analyses were done on a moderate sample 241 size and, as in any selected study population, results may not be directly extrapolated to 242 all people with Long COVID. Can authors discuss that why sample size was small? Is large population dataset not available? It should be clarified, and including large dataset will make their conclusions more robust. 

In Figure 1, labels must be in bold, so that it is more readable. 

Section 3.2, please indicate the cluster names when discussing about different clusters, which are now just named as 'Clusters-' in the text. Further looking at Figure S2, it is not clear that how authors determine cluster size as 3.. Can't it be 5? Please explain clearly, and this figure can go in the main text, and the manuscript discusses a lot about different clusters in main text, and then citing the related figure in supplementary. Further, please improve the quality of Figure S2. Labels are not clear, and figure description is poor. 

Author Response

Point 1: It is an interesting report highlighting "Long COVID" symptoms. However, as indicated by the authors that The analyses were done on a moderate sample 241 size and, as in any selected study population, results may not be directly extrapolated to 242 all people with Long COVID. Can authors discuss that why sample size was small? Is large population dataset not available? It should be clarified, and including large dataset will make their conclusions more robust. 

Our response to point 1: We thank Reviewer 2 for this comment. The sample size is limited for different reasons: first, the data were collected from participants from a well-characterized prospective cohort in Luxembourg. All participants had to have a PCR-confirmed initial COVID-19 infection in Luxembourg which limits the recruitment potential due to the small size of the country. Secondly, participants were followed longitudinally and completed the questionnaires on a voluntary basis, and some participants did not complete the 12-month questionnaire. Finally, we kept in our analysis only participants with a fully completed questionnaire and declaring at least 1 symptom at 12 months to ensure the quality of our analysis. However, we agree with Reviewer 2 that our results would benefit to be validated in a large dataset, this mentioned in the Discussion, lines 253-254 

Point 2: In Figure 1, labels must be in bold, so that it is more readable. 

Our response to point 2: Thank you for raising this visualization issue. Figure 1 has been improved so that labels are more easily readable.

Point 3: Section 3.2, please indicate the cluster names when discussing about different clusters, which are now just named as 'Clusters-' in the text. 

Our response to point 3: Thank you for this remark. After double checking the cluster names were already in the text but with additional spaces. We have now corrected the display of cluster names in the manuscript.

Point 4: Further looking at Figure S2, it is not clear that how authors determine cluster size as 3.. Can't it be 5? Please explain clearly, and this figure can go in the main text, and the manuscript discusses a lot about different clusters in main text, and then citing the related figure in supplementary. Further, please improve the quality of Figure S2. Labels are not clear, and figure description is poor.

Our response to point 4: We thank Reviewer 2 for this comment. The choice of cluster numbers is indeed a critical point when applying clustering methods. With the elbow method that is used in our manuscript, the optimal number of clusters is determined by the inflection point on the curve which corresponds in our case to 3. However, we also tested 4 and 5 as cluster numbers but this resulted in clusters with too few participants to be relevant. We improved the quality of the figure and included it in the main text as “Figure 1”.